Isolation and characterization of a novel lytic bacteriophage Pv27 with biocontrol potential against Vibrio parahaemolyticus infections in shrimp

Hien Vu Thi 1 2
Lanh Pham Thi 1
Pham Thao Thi Phuong 3
Tran Khang Nam 1 2
Duy Nguyen Dinh 1
Hoa Nguyen Thi 1
Canh Nguyen Xuan 4
Nguyen Quang Huy 2
http://orcid.org/0000-0003-3465-7118 Kim Seil 5 6
http://orcid.org/0000-0003-1002-7517 Quyen Dong Van 1 7 dvquyen@gmail.com
1 Laboratory of Molecular Microbiology, Institute of Biotechnology, Vietnam Academy of Science and Technology , Hanoi , Vietnam
2 University of Science and Technology of Hanoi, Vietnam Academy of Science and Technology , Hanoi , Vietnam
3 Vinmec Healthcare System , Hanoi , Vietnam
4 Vietnam National University of Agriculture , Hanoi , Vietnam
5 University of Science & Technology (UST) , Daejeon , Republic of South Korea
6 Korea Research Institute of Standards and Science , Daejeon , Republic of South Korea
7 Graduate University of Science and Technology, Vietnam Academy of Science and Technology , Hanoi , Vietnam
Breitbart Mya
Electronic publication date: 2025 May 6
Publication date: 2025
Volume: 13
Electronic Location ID: e19421
Received 2025 Feb 21; Accepted 2025 Apr 14
Copyright: © 2025 Hien et al.
Copyright year: 2025
Copyright holder: Hien et al.
License: This is an open access article distributed under the terms of the Creative Commons Attribution License, which permits unrestricted use, distribution, reproduction and adaptation in any medium and for any purpose provided that it is properly attributed. For attribution, the original author(s), title, publication source (PeerJ) and either DOI or URL of the article must be cited.
License URL: https://creativecommons.org/licenses/by/4.0/

Keywords: Vibrio parahaemolyticus, Shrimp, Bacteriophages, Biocontrol, Transmission electron microscopy (TEM), Host range, Aquaculture, Acute hepatopancreatic necrosis disease (AHPND), Phage therapy

Funding: Vietnam Academy of Science and Technology CT0000.01/22-24 This study was supported by the Vietnam Academy of Science and Technology, under grant No. CT0000.01/22-24. The funders had no role in study design, data collection and analysis, decision to publish, or preparation of the manuscript.

==============================
Background

Vibrio parahaemolyticus is a major disease-causing species of Vibrio that is pathogenic to both farmed shrimp and humans. With the increasing spread of antibiotic-resistant V. parahaemolyticus strains, bacteriophages (or phages) are considered potential agents for biocontrol as an alternative to antibiotics. In this study, a bacteriophage capable of lysing V. parahaemolyticus, named Pv27, was isolated, characterized, and evaluated for its potential to control Vibrio infections as a natural therapy.

Methods

Phage Pv27 was isolated using the double-layer agar technique and its morphology was characterized by transmission electron microscopy (TEM). We further assessed the host range specificity, optimal multiplicity of infection (MOI), one-step growth kinetics, and environmental stability of Pv27 under various pH and temperature conditions. The inhibitory activity of Pv27 against V. parahaemolyticus was evaluated in vitro. Finally, genomic analysis of Pv27 was conducted through whole-genome sequencing, followed by functional annotation of open reading frames (ORFs) and phylogenetic analysis.

Results

Phage Pv27 exhibited a Myovirus-like morphology, characterized by an icosahedral head (92.7 ± 2 nm) and a contractile tail (103 ± 11 nm), and belongs to the class Caudoviricetes. Pv27 demonstrated high lytic activity against its host cells, with a short latent period of approximately 25 minutes and a large burst size of 112 plaque-forming units (PFU) per infected cell. The phage displayed significant tolerance to a wide pH range (from 3 to 11) and remained heat-stable at temperatures up to 60 °C for 90 min. Genetically, Pv27 possesses a circular double-stranded DNA genome spanning 191,395 base pairs, with a G + C content of 35% and comprising 355 open reading frames (ORFs). Remarkably, up to 23 tRNA genes were identified in its genome, while no genes associated with antibiotic resistance, virulence, or lysogeny were detected, suggesting its potential as a valuable biocontrol agent. Results from the VIRIDIC, Basic Local Alignment Search Tool (BLAST) and phylogenetic analyses revealed that Pv27 is closely related to the two known Vibrio phages, phiKT1024 and phiTY18. Several genes associated with enhanced environmental competitiveness were also identified in the Pv27 genome, including those encoding a PhoH-like phosphate starvation-inducible protein and endolysin. Phage Pv27 effectively lyses V. parahaemolyticus highlighting its potential as a biocontrol agent.

Introduction

Vibrio parahaemolyticus, a Gram-negative, halophilic bacterium, naturally inhabits estuarine and marine environments, commonly occurring in aquaculture settings and posing a significant threat to both aquatic life and human health (Chen et al., 2023; Liang et al., 2022). V. parahaemolyticus is a pathogenic marine bacterium that can cause acute hepatopancreatic necrosis disease (AHPND) in farmed shrimp, leading to significant economic losses in shrimp aquaculture (Flegel, 2012; Jun et al., 2016). Shrimp affected by AHPND exhibit lethargy, anorexia, slow growth, and an empty digestive tract, with the disease characterized by severe atrophy of the hepatopancreas and a mortality rate of up to 100% within the first 35 days of the post-larval stage (Hong, Lu & Xu, 2016). V. parahaemolyticus is widely recognized for causing AHPND in shrimp via toxin production (pirA, pirB) (Han et al., 2015). However, some strains cause disease through non-toxin mechanisms like adhesion, biofilm formation, and extracellular enzyme secretion, leading to severe outbreaks and economic losses (Kumar et al., 2014; Zhang et al., 2021). In addition, V. parahaemolyticus poses a threat to food safety, triggering severe diarrhea and acute gastroenteritis in human globally (Yang et al., 2019). In low- and middle-income countries, the estimated cost of food-borne diseases were over US$100 billion per year (Grace, 2023). In human, foodborne illnesses related to V. parahaemolyticus often result from consuming contaminated food carrying bacteria and toxins like thermostable direct hemolysin (TDH), TDH-related hemolysin (TRH), and type III secretion systems 1 (T3SS1) and 2 (T3SS2) (Beshiru & Igbinosa, 2023; Wang et al., 2015). Infected individuals typically develop symptoms such as diarrhea, vomiting, abdominal cramps, nausea, fever, and stomach pain after 24 h (Rezny & Evans, 2024).

Antibiotics are widely used in commercial hatchery operations to control the infection of V. parahaemolyticus, thereby rising a risk of antibiotic-resistant bacterial strains (Elmahdi, DaSilva & Parveen, 2016). Many V. parahaemolyticus strains displayed resistance to commonly used antibiotics, i.e., ampicillin, erythromycin, cefazolin, and especially streptomycin (Kang et al., 2017). Thus, the spread of antibiotic-resistant strains not only poses a major challenge to public health and economies but also undermines the effectiveness of antibiotic-based treatments.

The proliferation of antibiotic-resistant strains of V. parahaemolyticus further complicated its management, undermining the efficacy of traditional antibiotic treatments and necessitating the exploration of alternative strategies to control its spread and impact (Ioannou, Baliou & Samonis, 2023). Bacteriophages or phages, are viruses that exclusively infect bacterial cells, and capable of controlling many infectious diseases in plants, animals and humans (Chhibber, Kaur & Kumari, 2008; de Sousa et al., 2023; Jun et al., 2012; Sarker et al., 2012). The application of phage therapy in aquaculture offers a natural and targeted method to mitigate bacterial infections without the adverse environmental impacts often associated with antibiotic use. Moreover, phages exhibit high specificity, targeting particular bacterial species or strains which helps minimize their impact on commensal bacteria (Tan et al., 2021). Therefore, bacteriophages have been studied as alternative biocontrol agents (Aldayel, 2019). Numerous studies have focused on identifying, characterizing, and applying Vibrio phages in animals. To date, several bacteriophages have been isolated and were evaluated for their application against V. parahaemolyticus infection in shrimp, such as phage phiTY18 (Liu et al., 2022), phage vB_ValM_PVA8 (Fu et al., 2023), phage SSJ01 (Kang & Chang, 2024), phage vB_VpaP_GHSM17 (Liang et al., 2022), phage vB_VpaS_PGA and vB_VpaS_PGB (Zeng et al., 2024), phage KIT05 (Anh et al., 2022), phage R01 (Li et al., 2023). In addition, recent studies have also demonstrated the potential of bacteriophage in reducing bacterial loads and biofilm formation in settings, such as seafood surfaces and aquaculture environments using phage BPVP-3325, CAU_VPP01, and R16F (Chen et al., 2023; Jang et al., 2022; Kim et al., 2024). These outcomes have proven the efficacy of phage therapy in practical applications. Since the lytic activity of phage is typically host-specificity, it is essential to isolate and screen phages that target bacterial pathogens prevalence in the local or national environment.

In this study, we aim to isolate and characterize the biological features of a bacteriophage possessing the ability to lyse V. parahaemolyticus in Vietnam and evaluate its potential as a biocontrol agent. Our findings provide the materials as well as fundamental knowledge about the applicability of phage therapy to combat V. parahaemolyticus infections in shrimp farming, promoting the sustainable development of aquaculture.

Materials and Methods

Sampling

Water and sediment samples were collected from shrimp ponds with shrimp showing clinical signs of Vibrio disease (hepatopancreas atrophy, loss of colour, black spots or streaks, empty stomach and midgut, and soft bodies) located in Quang Yen district, Quang Ninh province, Vietnam (20°55′40″N 106°51′5″E).

Bacterial strains

Twenty bacterial strains of Vibrio spp. (V. parahaemolyticus, Vibrio harveyi, Vibrio alginolyticus, Vibrio cholera) isolated from diseased shrimp and other bacterial species, including Staphylococus aureus, Bacillus cereus, Bacillus subtilis and Lactobacillus plantarum (Table 1) were obtained from the bacterial collection at the Laboratory of Molecular Microbiology, Institute of Biotechnology, Vietnam Academy of Science and Technology.

Table 1 Host range of phage Pv27.

Bacterial species	Strain	Source	Location	Virulence genes	Plaque formation	
Vibrio parahaemolyticus	Vp-ATCC 17802	American type culture collection		toxR, tlh, trh	–	
Vp-NA1	Dead shrimp	Quynh Luu, Nghe An, Vietnam	toxR, tlh, PirA	–	
Vp-NA2	Shrimp ponds	Quynh Luu, Nghe An, Vietnam	toxR, tlh	–	
Vp-HP1	Diseased shrimp	Kien Thuy, Hai Phong, Vietnam	toxR, tlh, tdh, PirA	++	
Vp-HP2	Diseased shrimp	Kien Thuy, Hai Phong, Vietnam	toxR, tlh	–	
Vp-QN1	Dead shrimp	Quang Yen, Quang Ninh, Vietnam	toxR, tlh, tdh	–	
Vp-QN2	Diseased shrimp	Quang Yen, Quang Ninh, Vietnam	toxR, tlh, tdh	–	
Vp-QN3	Shrimp ponds	Quang Yen, Quang Ninh, Vietnam	toxR, tlh	+	
Vp-QN4	Shrimp ponds	Quang Yen, Quang Ninh, Vietnam	toxR, tlh	–	
Vp-MT1	Dead shrimp	Bac Lieu, Vietnam	toxR, tlh	+	
Vp-MT2	Shrimp ponds	Vinh Trach Dong, Bac Lieu, Vietnam	toxR, tlh, trh	–	
Vp-MT3	Shrimp ponds	Vinh Trach Dong, Bac Lieu, Vietnam	toxR, tlh, tdh	–	
Vibrio alginolyticus	Va-QN1	Shrimp ponds	Quang Yen, Quang Ninh, Vietnam	N/A	–	
Va-MT2	Shrimp ponds	Vinh Trach Dong, Bac Lieu, Vietnam	N/A	–	
Va-MT10	Shrimp ponds	Vinh Trach Dong, Bac Lieu, Vietnam	N/A	–	
Va-MT21	Shrimp ponds	Vinh Trach Dong, Bac Lieu, Vietnam	N/A	–	
Vibrio harveyi	Vh-MT23	Shrimp ponds	Vinh Trach Dong, Bac Lieu, Vietnam	N/A	–	
Vh-MT34	Shrimp ponds	Vinh Trach Dong, Bac Lieu, Vietnam	N/A	–	
Vh-MT38	Shrimp ponds	Vinh Trach Dong, Bac Lieu, Vietnam	N/A	–	
Vibrio cholerae	Vc-0103	Shrimp ponds	Vinh Trach Dong, Bac Lieu, Vietnam	N/A	–	
Staphylococcus aureus	Sa816	N/A	N/A	N/A	–	
Lactobacillus plantarum	Lp-03	N/A	N/A	N/A	–	
Bacillus cereus	VTCC 11265	N/A	N/A	N/A	–	
Bacillus subtilis	Bs-01	N/A	N/A	N/A	–	
Note:

Different bacterial species and strains were used as the host to determine the host spectrum of the phage using a spot test, (++: clear plaques; +: slightly turbidity plaques; –: no plaques formed; N/A: Not available).

Bacteriophage isolation

Bacteriophages were isolated from the collected water and sediment samples with V. parahaemolyticus Vp-HP1 (Table 1) as a host using the method previously described by Liang et al. (2022), with slight modifications (Liang et al., 2022). Briefly, 5 g of mud was mixed with 50 mL of pond water, and 50 mL of the supernatant was added into 200 mL of the host bacterial culture at the mid-log phase. The mixture was incubated at 37 °C with shaking overnight, followed by centrifugating at 10,000 rpm for 10 min at 4 °C. Cell pellets and large particulates were discarded, and the supernatant was collected and filtered using a 0.22-µm filter (Sartorius, Germany). The presence of phage was detected by the formation of plaques on a double agar plate using a double agar spot assay (Abedon, 2018; Ács, Gambino & Brøndsted, 2020) with a host bacterial strain. A total of 10 µL of phage suspension was added directly onto the soft agar containing the host, incubated at 37 °C overnight to observe plaque formation. The isolated phage was purified by consecutive single-plaque isolation and subsequently enriched. The phage titer was determined by serial dilution (101–107) in SM buffer (NaCl 100 mM, MgSO4.7H2O 8 mM, Tris-HCl 50 mM pH 7.5 and gelatin 0.01%). The diluted phage solution was mixed with the bacterial culture and added in 5 mL of LB soft agar (0.7%), then poured onto LB agar (2%) plate for incubation at 37 °C overnight. The purified phage was stored in SM buffer at 4 °C for further experiments.

Transmission electron microscopy

The morphology of phage Pv27 was observed using the transmission electron microscopy (TEM) JEM-1400 Flash (Jeol, Japan). The purified phage suspension (1010 PFU/mL) was absorbed onto a copper grid for 10 min. The grid was fixed in uranyl acetate 2% (w/v) for 2 min and dried at room temperature. The phage morphology was observed using TEM at an accelerating voltage of 80 kV (Ackermann, 2009). It was classified and identified based on Committee on Taxonomy of Viruses (ICTV) guidelines.

Host range assay

A total of 20 Vibrio strains (including 12 strains of V. parahaemolyticus, three V. haveyii strains, four V. alginolyticus strains, one V. cholera strain), as well as one S. aureus strain, one B. cereus strain, one B. subtilis strain and one L. plantarum strain were used to determine the lytic host range of Pv27 using the drop test method (Table 1). In brief, 100 µl of V. parahaemolyticus in mid-log phase were mixed with 5 ml of warm soft agar (LB containing 0.7% agar) and poured onto solid LB agar plates (2% agar). After solidifying the soft agar for 10 min, 20 µl of phage solution was spotted onto the surface of the plates. The plates were incubated overnight at 37 °C, and the formation of lysis zones was observed. Host range of the phage was evaluated based on the clarity of lysis zones: clear lysis zone (+) and no lysis (−).

Multiplicity of infection

Phage Pv27 and its host were mixed at multiplicity of infection (MOI) in the range of 0.001, 0.01, 0.1, 1, 10 and 100. After incubation at 37 °C for 10 h, the mixture was centrifuged at 10,000 rpm for 5 min for cell removal. The supernatant was filtered through a 0.22-µm filter. The phage titer was determined using double-layer agar method. The optimal MOI is the ratio that generates the highest phage titer (Liang et al., 2022).

One-step growth curve

The one-step growth curve of phage Pv27 performed at MOI 0.001 was constructed using V. parahaemolyticus Vp-HP1 as a host. The latent time and burst size of Pv27 was determined as described previously with some modifications (Yang et al., 2020). The inoculum of host strain at log-phase (adjusted to 108 CFU/mL) was mixed with phage at MOI of 0.001. The mixture was incubated at 37 °C for 10 min to allow phage absorption, then centrifuged at 10,000 rpm for 5 min to remove free phages. The pellets were resuspended in 10 mL of LB broth and incubated at 37 °C with shaking. Samples were collected every 5 min for the first 30 min, then every 10 min during the later period, to determine the phage titer. The experiment was conducted in triplicate. The burst size was calculated as the ratio of the number of phages formed during the rise period to the estimated number of infected cells (Nabergoj, Modic & Podgornik, 2017).

Stability of phage Pv27

The stability of Pv27 toward the effects of temperature, pH, and salinity on phage survival rate were assessed as previously described with some modifications (Tan et al., 2021; Yang et al., 2020). To evaluate the effect of temperature on phage viability, the phage in SM buffer was incubated at 4 °C, 20 °C, 37 °C, 50 °C, 60 °C and 70 °C for 90 min. To determine the impact of pH on Pv27 stability, the phage was incubated in buffers of different pH ranging from 2 to 11 and the phage in SM buffer (pH 7) was used as a control. To investigate the effect of salinity, sterile synthetic seawater was prepared with a final salinity of 1.5%, 3%, 5%, and 10%, and the phage was added to each solution. During the experiments, the samples were kept at room temperature for 90 min. Each trial was carried out in triplicate as described previously (Yin et al., 2019). After incubation, phage titer was determined using the double-agar technique.

In vitro bacteriolytic assay

The bacteriolytic activity of the phage at different MOIs was determined by Chen’s method with some modifications (Chen et al., 2023) using V. parahaemolyticus Vp-HP1 as a host. The exponential-phase culture of the host strain was added to the 96-well plate and was impregnated with the phage suspension at different MOIs (0.001, 0.1, 1, 10, and 100). Wells incubated with bacteria or phage only served as controls. Subsequently, the mixture was incubated at 37 °C with orbital shaking for 24 h. Samples were collected at every one hour and measured using optical densitometry (Eppendorf Bio Photometer plus, Hamburg, Germany) at 600 nm. The experiment was done in triplicate.

DNA extraction and genome sequencing

Genomic DNA was extracted from the phage particles using a modified phenol-chloroform-isoamyl alcohol method (Borges, 2022; Orozco-Ochoa et al., 2023). Briefly, 1 mL of phage suspension containing 1010 PFU/mL was treated with DNase I and RNase A at the final concentrations of 0.8 U/mL and 0.1 mg/mL, respectively. The mixture was then incubated for 1 h at 37 °C to remove bacterial DNA contamination. Purified phages were treated with proteinase K, EDTA, and SDS at the final concentrations of 20 mM, 0.5 mg/ml, and 0.5%, respectively, at 55 °C for 1 h to digest the phage capsid. The DNA phage was purified with an equal volume of phenol/chloroform/isoamyl alcohol (25:24:1) and chloroform/isoamyl alcohol (24:1), then centrifuged at 12,000 g for 20 min. The aqueous layer was collected, mixed with an equal volume of isopropanol, and kept at −20 °C for 1 h to precipitate the total DNA. The mixture was then centrifuged at 12,000 g for 20 min at 4 °C, the DNA pellets were washed twice with 75% ethanol. Lastly, the DNA was air dried at room temperature, resuspended in DNase/Rnase-free deionized water, and stored at −20 °C. DNA quality and quantity were determined by NanoDrop 2000 (Thermo Fisher Scientific, Waltham, MA, USA) and sequenced by the Illumina HiSeq platform (Illumina, San Diego, CA, USA).

Bioinformatics analysis

The raw paired-end reads were processed by fastp (version 0.20.0) with default parameter (Chen, 2023). De novo assembly was performed using the SPAdes pipeline (Bankevich et al., 2012) and assembly quality was checked by QUAST (Gurevich et al., 2013). Open reading frames (ORFs) were predicted by Pharokka (Bouras et al., 2023). Annotated proteins were checked by the Basic Local Alignment Search Tool (BLAST) search against UniprotKB reference proteomes + Swissprot database to interrogate functional category determination by Pharokka. Circular genome mapping was plotted by the CGView online server (https://cgview.ca/). The presence of antibiotic resistance genes was detected by the RGI online server (https://card.mcmaster.ca/analyze/rgi) with the Comprehensive Antibiotic Resistance Database (CARD), and virulence determinants were checked against the Virulence factor database (VFDB) (http://www.mgc.ac.cn/VFs/search_VFs.htm). The similarity analysis of phage genome nucleotide sequences was conducted using VIRIDIC (Virus Intergenomic Distance Calculator) v1.0 (http://rhea.icbm.uni-oldenburg.de/VIRIDIC/) (Moraru, Varsani & Kropinski, 2020). The Viral Proteomic Tree server (VipTree) was used to analyze genome-wide sequence similarities computed by tBLASTx and closed phage genomes were selected for generating genome comparison figures (Nishimura et al., 2017). The phylogenetic trees based on two proteins, terminase large subunit and major capsid protein, of phage Pv27 and other related phages were constructed by MegaX v11.0 using the maximum likehood with 1,000 bootstrap replicates (Kumar et al., 2018). To further investigate the taxonomy relationship of Vibrio phage Pv27, gene-sharing network-based approaches were conducted by aligning whole genome sequences with reference phages. Phage pv27, along with reference genomes downloaded from the INfrastructure for a PHAge REference Database (INPHARED) in September 2024 (Cook et al., 2021) were run through BLASTp all vs. all mode. Gene-sharing network classification was then performed using vConTACT2 (v0.9.22). To assign taxonomy based on similarities to reference viral groups, a phage proteomic tree was built by Virus Classification and Tree Building Online Resource (VICTOR) (Meier-Kolthoff & Göker, 2017). The genome-blast distance phylogeny (GBDP) method was used to conduct pairwise comparisons of the nucleotide sequences (Meier-Kolthoff et al., 2013).

Statistical analysis

The data of phage titers were analyzed and visualized by using IBM SPSS Statistics software version 20 (IBM Corp., Armonk, NY, USA). The statistical differences in phage titers among the experimental groups were determined using one-way ANOVA and Tukey’s post-hoc test. Results are presented as mean ± SD, with statistical significance at p < 0.05.

Results

Phage isolation and morphology

A phage, namely Pv27 was isolated using double-layer agar technique. Clear plaques (about 1 mm in diameter) were formed on plates 20 h post-infection (Fig. 1A). TEM observations revealed that phage Pv27 has an icosahedral head with a diameter of 92.7 ± 2 nm, along with a narrow, short neck or collar region and a contractile tail of 103 ± 11 nm in length (Fig. 1B). According to the guidelines of the International Committee on Virus Classification (ICTV) in 2022, phage Pv27 belongs to Myovirus-like morphology, class Caudoviricetes.

Figure 1 Characterization of Pv27.

(A) Plaque appearance. (B) Virion morphology.

Host range of the phage

Pv27 exhibited high host-specificity against V. parahaemolyticus strains. They lysed three out of the 13 V. parahaemolyticus (Vp-HP1, Vp-QN3 and Vp-MT1) and showed no inhibitory activity against the remaining V. parahaemolyticus strains tested (V. parahaemolyticus ATCC 17802 and other 10 V. parahaemolyticus isolates), as well as other Vibrio species, including V. alginolyticus, V. harveyi, V. cholerae, and other bacterial species, including S. aureus, B. cereus, B. subtilis and L. plantarum (Table 1).

Optimal multiplicity of infection

Multiplicity of infection (MOI) refers to the ratio of phage particles to host cells. To optimize the yield of bacteriophage production, it is essential to determine the MOI. The results showed that high phage titers were produced at different MOIs of 0.001, 0.01, 0.1, 1, 10, and 100, with the highest phage titer of 11.64 log PFU/mL obtained at the MOI of 0.001 (p < 0.05). Therefore, the MOI of 0.001 was considered the optimal MOI for phage Pv27 (Fig. 2B) and was used in subsequent experiments.

Figure 2 Biological characterization of Pv27.

(A) One-step growth curve. (B) MOI. (C) Thermal stability. (D) pH stability. (E) Salinity stability. All data are shown as the means of three independent experiments. Error bars represent the standard deviation (SD) of the Mean. Different letters (a–e) indicate statistically significant differences (p < 0.05).

One-step growth curve

To characterize the replication dynamics during the infection cycle of Pv27, we performed one-step growth curve analysis. The results showed that phage Pv27 has a short latent period of approximately 25 min (Fig. 2A). Subsequently, the phage entered the burst phase, and the phage titer was steadily raised in the next 35 min and reached a stationary phase after 60 min at 8.9 log PFU/mL (Fig. 2A). The burst size was determined to be 112 PFU per infected cell.

Thermal, pH and salinity stability

The stability of phage Pv27 was assessed under various thermal conditions, and pH and salinity levels. Our results revealed that phage Pv27 retained its activity at temperatures below 50 °C. At 60 °C, the phage titer decreased approximately 2.5 log PFU/mL after 90 min of incubation, but complete inactivation occurred at 70 °C (Fig. 2C). Similarly, phage Pv27 demonstrated stability across a wide pH range (5–11), as indicated by consistently maintained phage titers. However, exposure to pH 3 resulted in a significant reduction in phage titer of 1.4 log PFU/mL (p < 0.05), and the phage was completely inactivated at extreme pH values at 2 and 12 (Fig. 2D). The salinity stability test demonstrated that salinities ranging from 1.5% to 10% did not affect the titer of phage Pv27 in 90 min (p > 0.05) (Fig. 2E).

In vitro lytic activity of phage Pv27

The results demonstrated that phage Pv27 significantly inhibited the growth of the tested bacterium and delayed its progression to the stationary phase compared to the control group (p < 0.05) (Fig. 3). In the control group, the bacterial cell density (measured as OD600) gradually increased over the experimental period, reaching the stationary phase after 12 h of culture. In contrast, bacterial growth in all phage Pv27-treated groups was effectively inhibited during the first eight hours. Although recovery occurred subsequently, the stationary phase was reached at a lower OD600 value (approximately 0.6–0.8) after 20 h of treatment. Among the evaluated multiplicities of infection (MOIs), MOIs of 0.001, 10, and 100 were the most effective, exhibiting lower OD600 values at most time points, particularly during the stationary phase (Fig. 3).

Figure 3 Lytic activity of phage Pv27 against V. parahaemolyticus Vp-HP1 at five different MOIs.

The absorbance (OD600) read indicates bacterial growth.

General features of phage Pv27 genome

Genomic analysis revealed that phage Pv27 possesses a circular double-stranded DNA genome (Fig. 4) comprising 191,395 bp with a G + C content of 35%. The genome sequence of phage Pv27 has been deposited in GenBank under the accession number OR413349. Comparative genomic analysis using BLASTn showed that the genome of phage Pv27 shares the highest nucleotide sequence identity of 98.33% (with 96% query coverage) with Vibrio phage phiKT1024 (OM249648.1) and 92.23% (with 92% query coverage) with Vibrio phage Val (MK387337.2). A total of 355 ORFs were predicted, of which 88 ORFs (24.8%) were identified as encoding functional proteins, while the remaining ORFs (75.2%) were annotated as hypothetical proteins with unknown functions.

Figure 4 The genome map of bacteriophage Pv27.

From inwards to onwards, the circle represents GC content (black), GC skew (green indicates GC skew+ and purple indicates GC skew-), ORFs (clockwise indicates the forward reading frame and counterclockwise indicates the reverse reading frame).

All 88 predicted ORFs were categorized into six main functional modules: (1) DNA, RNA, and nucleotide metabolism; (2) structural proteins; (3) head and packaging proteins; (4) host lysis; (5) moron, auxiliary metabolic genes, and host takeover (AMG); and (6) other phage-related proteins (Table S1). Module (1) consists of 33 ORFs encoding DNA helicase (ORF345), DNA primase (ORF318), exonuclease and the endonuclease (ORF146, ORF316, ORF323, ORF58, ORF62 and ORF99), endonuclease VII (ORF 51), HNH endonuclease (ORF 62), anaerobic ribonucleoside reductase large and small subunits (ORF 134 and ORF 135), RNA ligase (ORF 233), two DNA polymerases (ORF 210 and ORF 212), among others (Table S1). Notably, ORF345 encoding DNA helicase, showed 97.9% nucleotide sequence identity to that of Vibrio phage Val (MK387337.2). Similarly, ORF318 encoding DNA primase, exhibited 99.7% identity to that of Vibrio phage phiKT1024 (OM249648.1). Additionally, all six ORFs encoding enzymes involved in nucleotide metabolism (exonucleases and endonucleases) displayed over 98% sequence similarity to the corresponding genes of Vibrio phage phiKT1024 (OM249648.1).

In module 2, the ORFs were further categorized into connector (three ORFs) and tail-associated genes (15 ORFs). Identified genes included four tail tube proteins (ORF 25, 26, 27 and 53), two tail sheaths (ORF 23 and ORF 288), two baseplate proteins (ORF 41 and ORF49), a baseplate tail tube cap (ORF43), a tail fiber protein (ORF287), a baseplate hub (ORF40), a baseplate hub subunit and a tail lysozyme (ORF46), a RNA ligase and tail fiber protein attachment catalyst (ORF257), a baseplate wedge subunit (ORF48), a long tail fiber protein distal subunit (ORF286). Additionally, three head-tail connectors were identified, including head-tail adaptor Ad2 (ORF18), head closure (ORF19) and head closure Hc2 (ORF20).

Meanwhile, module 3 included large terminase (TerL) (ORF245), head maturation protease (ORF 31, 118, 177), head scaffolding protein (ORF32), portal protein (ORF21), virion structural protein (ORF5) and major head protein (ORF33). This major head protein belongs to Gp23/Gp24_T4-like bacteriophage-like family. In module 4, four genes involved in host cell wall lysis were identified, including two endolysins (ORF42 and ORF82) and two Rz-like spanins (ORF174 and ORF175). ORF42 showed 99.89% identify with the amino acid sequence of the predicted peptidase family M15 of Vibrio phage phiKT1024 (OM249648.1). ORF82 shared 97.92% identity with the amino acid sequence of the predicted chitinase of Vibrio phage phiKT1024 (OM249648.1). This chitinase is an endolysin with lysozyme activity and belongs to the glycoside hydrolase family 19. The classification of protein families based on InterProScan results indicated that ORF174 has the structure of i-spanin, which includes a transmembrane region, a cytoplasmic domain, a non-cytoplasmic domain, and a coiled region. Additionally, ORF175 exhibits the structure of o-spanin, featuring a prokaryotic lipoprotein domain, a signal peptide region, and a non-cytoplasmic domain.

In module 5, three AMGs were identified, including an ABC transporter (ORF71), a membrane protein (ORF187), and a Lar-like restriction alleviation protein (ORF260). Furthermore, several ORFs encoding other functions were also identified, such as deoxynucleoside monophosphate kinase (ORF3), co-chaperonin GroES (ORF14), chaperonin GroEL (ORF24), phosphate starvation-inducible protein PhoH-like (ORF299), and beta-glucosyl-HMC-alpha-glucosyltransferase (ORF329). Additionally, 23 tRNAs were identified. Remarkably, no genes encoding antibiotic resistance, virulence factors, or lysogeny-related proteins (e.g., integrase) were found in the genome of phage Pv27.

Phylogenetic analysis

The VIRIDIC analysis showed intergenomic similarity between phage Pv27, related Vibrio phages and other phages (Fig. 5). The intergenomic similarity of Pv27 from the current study and Vibrio phage phiKT1024 (Accession no. OM249648) was 95.1%, indicating that these phages belong to the same species. Additionally, VIRIDIC analysis showed that Pv27 exhibited 82% to 94.7% intergenomic similarity with Va1 (Accession no. MK387337), VB_VaC_TDDLMA (Accession no. PP083315), VB_VaC_SRILMA (Accession no. PP083314), and phiTY18 (Accession no. MW451250), suggesting that they belong to the same genus based on the default VIRIDIC similarity thresholds of 70% for genus classification and 95% for species classification. However, all these Vibrio phages are currently classified under the unclassified Vibrio phage family within the class Caudoviricetes, according to the 2022 taxonomy update by the ICTV Bacterial Viruses Subcommittee (Turner et al., 2023), despite having previously been assigned to the Myoviridae family within the order Caudovirales. Similarly, the top BLASTN matches in the NCBI database revealed that Pv27 shared the highest homology with five bacteriophages previously reported, including Vibrio phage phiKT1024, phiTY18, Va1, VB_VaC_TDDLMA, and VB_VaC_SRILMA. Genomic collinearity analysis highlighted the similarity among these phages (Fig. S1).

Figure 5 VIRIDIC heatmap of the phage Vp27.

Intergenomic similarities of the phages are shown on the right side with the colored scale. The aligned genome fraction and genome length ratio values are shown on the left side with their corresponding scale.

The major capsid protein and terminase large subunit amino acid sequences of phage Pv27 and other related phages were used to construct phylogenetic trees. As a result, the two Maximum Likelihood phylogenetic trees obtained all clustered Pv27 with five other Vibrio phages, including phiKT1024, phiTY18, Va1, VB_VaC_TDDLMA, and VB_VaC_SRILMA into the same clade, which was defined as an unclassified Vibrio phage family (Caudoviricetes class with a Myovirus morphotype) (Figs. S2 and S3).

Further comparative genomic analysis using vContact2 revealed that phage Pv27 and related phages are in subcluster VC_0_0 within the VC_0 cluster, which includes Kyanoviridae, Straboviridae, and an unclassified Campylobacter phage family (Fig. 6). This analysis places Pv27 in VC_0_0, showing distant genetic links to Straboviridae and the unclassified group, suggesting that Pv27 may belong to an unclassified family or genus. Additionally, the viral proteomic tree constructed using VICTOR classified phage Pv27 into a clade that includes phiKT1024 and phiTY18 (Fig. 7).

Figure 6 Protein-sharing network of Pv27 and known phages generated with vConTACT2.

Phages (nodes) are connected by edges, indicating the significant pairwise similarity between them in terms of shared protein contents. The sub-cluster (VC_0_0) of Pv27 and its related phages are enlarged inside cluster VC_0, which consists of members of the family Kyanoviridae, Straboviridae, and the unclassified family of Campylobacter phage.

Figure 7 Whole genome based proteomic tree of Vibrio phage Pv27 and selected sequence in VC_0_1 from NCBI database and constructed by Virus Classification and Tree Building Online Resource (VICTOR) with the formula d6.

The proteomic tree consists of 27 phage genomes and yields an average support of 77%. Three series of color boxes behind the tree indicate family, sub-family, and genera classified by OPTSIL.

Discussion

Phages are the most abundant life forms on Earth, characterized by their high specificity for bacterial hosts, which makes them valuable tools in various fields, including aquaculture (Reyneke et al., 2024; Rogovski et al., 2021). Their application in the biocontrol of pathogenic bacteria in the shrimp industry is particularly promising (Nachimuthu et al., 2021). Lytic phages infect bacterial cells, replicate using the host’s cellular machinery, and cause cell lysis, thereby releasing progeny into the surrounding environment (Dang & Sullivan, 2014). Bacteriophage isolation typically occurs in the natural habitats of their host bacteria, with success rates influenced by factors such as phage density, host activity, and the methods used for isolation (Greer, 2005). Previous studies have reported the isolation of lytic bacteriophages targeting Vibrio species from diverse environments, including seawater, river estuaries, market wastewater, and shrimp ponds (Alagappan et al., 2010; Hsu et al., 2024; Kim et al., 2024; Stalin & Srinivasan, 2016). In this study, we isolated and characterized the lytic phage Pv27, specific to V. parahaemolyticus infecting shrimp, from water and sediment samples collected from diseased shrimp ponds in Vietnam.

Previous studies have shown that most reported Vibrio phages belong to the order Caudovirales and exhibit diverse families and morphologies, including elongated heads with contractile tails (Fu et al., 2023); icosahedral capsids with short, non-contractile tails (Kang & Chang, 2024; Liang et al., 2022); icosahedral capsids with complete, contractile tails (Chen et al., 2023; Zeng et al., 2024; Liu et al., 2022); icosahedral symmetrical heads with very short tails (Anh et al., 2022); icosahedral head with a long and flexible tail (Brossard Stoos et al., 2022) and elongated capsids with long, non-flexible tails (Li et al., 2023). Moreover, a V. parahaemolyticus bacteriophage that belongs to family Inoviridae, class Faserviricetes, order Tubulavirales with filamentous shaped morphology was also recently reported (Dubey et al., 2021). In this study, TEM analysis revealed that phage Pv27 has an icosahedral head and a contractile tail, resembling previously reported phages with Myovirus-like morphology, such as vB_VpaM_R16F (Chen et al., 2023), vB_VpaS_PGA, vB_VpaS_PGB (Zeng et al., 2024), and phiTY18 (Liu et al., 2022). This structure likely plays a critical role in the infection process and facilitates the injection of the viral genome into bacterial cells (Liu et al., 2021).

Phage Pv27 exhibited a narrow host range, similar to recently reported Vibrio phages such as phiTY18 (Liu et al., 2022) and phage vB_VpS_PG28 (Tian et al., 2022). Our results revealed that phage Pv27 displayed high specificity toward V. parahaemolyticus strains, lysing only three out of 13 tested strains, with the highest lytic activity observed against the Vp-HP1 strain. However, Pv27 exhibited no lytic activity against other Vibrio species or non-Vibrio bacteria, including S. aureus, B. cereus, E. coli and beneficial bacteria such as B. subtilis and L. plantarum. In contrast, Fu et al. (2023) reported a broad-host-range bacteriophage, vB_ValM_PVA8 (PVA8), capable of efficiently infecting both pathogenic isolates of V. alginolyticus and V. parahaemolyticus, but not other bacterial species (Fu et al., 2023). Phage Pv27’s narrow host range is likely due to high receptor specificity, bacterial defense mechanisms, and co-evolutionary adaptation (Labrie, Samson & Moineau, 2010). Targeted action, such as that exhibited by Pv27, is advantageous for minimizing collateral effects on non-target bacteria, including beneficial microflora, which is a critical consideration for phage therapy applications (Tan et al., 2021). In addition, V. parahaemolyticus is a major pathogen in shrimp aquaculture, causing AHPND through the production of PirAB-like toxins. However, V. parahaemolyticus is also associated with other vibriosis infections that do not rely on toxin production but instead involve mechanisms such as adhesion, biofilm formation, and extracellular enzyme secretion. These non-toxin-mediated infections may be more amenable to bacteriophage-based control, as phages can effectively target and lyse bacterial populations without the need to specifically neutralize virulence genes. Expanding the use of phages beyond AHPND could provide a broader biocontrol strategy for managing V. parahaemolyticus infections in shrimp farming (Kumar et al., 2014; Zhang et al., 2021). Our results demonstrated the flexibility of phage therapy, as the isolated bacteriophage effectively lysed both the pirA toxin-producing strain (Vp-HP1) and two non-toxin-producing strains (Vp-QN3 and Vp-MT1). Future research could focus on phage cocktails, genetic modification, or natural selection strategies to expand Pv27’s host range and increase its applicability (Chan & Abedon, 2012; Pires et al., 2016).

The titer of Pv27 reached up to 1011 PFU/mL at an optimal MOI of 0.001. Moreover, one-step growth curve analysis revealed that phage Pv27 exhibits rapid replication and potent lytic activity, with a short latent period of approximately 25 min and a large burst size of 112 PFU per infected cell. This performance surpasses that of Vibrio phage phiTY18 (20 min and 48 PFU per infected cell) (Liu et al., 2022) and phage PGA (20 min and 88.4 PFU per infected cell) (Zeng et al., 2024). These findings suggest that after adsorption onto the host surface, phage Pv27 propagates effectively and rapidly. Furthermore, the in vitro assays demonstrated that phage Pv27 significantly inhibited the growth of V. parahaemolyticus and delayed its stationary phase, particularly at MOIs of 0.001, 10, and 100, compared to the control. These results indicate that phage Pv27 exhibits excellent lytic activity against V. parahaemolyticus.

For practical applications, bacteriophages must withstand diverse environmental conditions, as their thermal, pH and salinity stability directly impact their ability to target pathogens like V. parahaemolyticus in shrimp farming (Orozco-Ochoa et al., 2023; Yin et al., 2019). Temperature influences phage attachment, penetration, and reproduction, while elevated temperatures can cause irreversible damage or denaturation (Hu et al., 2021; Tan et al., 2021). Phage Pv27 demonstrated thermal stability up to 60 °C and retained infectivity across a broad pH range (pH 5–11). However, extreme conditions, such as temperatures exceeding 60 °C and pH levels outside the range of 3–11, significantly reduced its viability. These stability characteristics of Pv27 are comparable to those of Vibrio phage phiTY18 (Liu et al., 2022), which remains stable up to 50 °C and within a pH range of 5–9. The ability of Pv27 to tolerate a wide pH range is particularly critical for ensuring its viability and activity in aquatic environments with fluctuating pH levels. Besides, our study revealed that Pv27 remained stable across salinities ranging from 1.5% to 10%, suggesting its potential application in marine environments where V. parahaemolyticus is commonly detected at high levels of contamination as in a previous study (Yin et al., 2019).

Genomically, phage Pv27 possesses a double-stranded DNA genome comprising 191,395 bp with a G + C content of 35%. The result of BLASTn analysis showed that Pv27 had the highest nucleotide sequence identity of 98.33% (with 96% query coverage) with Vibrio phage phiKT1024 (OM249648.1) and 92.23% (with 92% query coverage) with Vibrio phage Val (MK387337.2). The Pv27 genome encodes 355 ORFs with a substantial proportion of hypothetical proteins, suggesting a complex genetic architecture with potential novel functionalities of phage Pv27. The presence of genes related to DNA metabolism, structural proteins, packaging proteins, and host lysis enzymes in the Pv27 genome underscores the advanced mechanisms utilized by phage Pv27 during infection and replication processes. Remarkably, most structural genes are located in the downstream region of phage Pv27 genome. Among these, fifteen tail-related proteins form a distinct cluster, including a tail lysozyme (ORF46), which plays a crucial role in tail assembly and facilitates phage penetration into the host cell’s outer membrane during infection (Yap & Rossmann, 2014). Comparative genomic analysis revealed that all tail-related proteins of Pv27 share over 98% similarity with those of Vibrio phage phiKT1024, suggesting that they both may infect similar bacterial hosts. This observation aligns with previous studies emphasizing the importance of structural modules, particularly tail-related proteins, in host recognition (Yang et al., 2022). Phage Pv27 genome also encodes enzymes critical for DNA replication and repair. Additionally, enzymes involved in nucleotide metabolism are present, which may contribute to the degradation of host genomic DNA and RNA to generate deoxyribonucleotides necessary for phage DNA synthesis (Gao et al., 2020). These findings highlight the multifaceted genetic toolkit of Pv27, enabling efficient genome packaging, structural assembly, replication, host recognition, and phage propagation.

Further analysis revealed that Pv27 encodes a terminase large subunit (ORF245), which functions as a key packaging molecule. This protein plays a crucial role in the genome packaging process, transferring viral DNA into the empty capsid and defining the start and end points of the packaging reaction (Ge et al., 2023). The portal protein, encoded by ORF21, is predicted to facilitate the injection of DNA into the host cell (Isidro, Henriques & Tavares, 2004). Additionally, it interacts with the terminase to regulate the size of the assembled viral genome and plays a critical role in phage maturation and infection (Ge et al., 2023). The major head protein of Pv27 is classified within the Gp23/Gp24_T4-like bacteriophage family, underscoring its structural and functional similarity to other T4-like phages. Together, these proteins play pivotal roles in the phage packaging mechanism, ensuring accurate encapsulation of the viral genome and enabling subsequent processes of replication and infection.

For most double-stranded DNA bacteriophages, bacterial lysis is mediated by phage-encoded muralytic enzymes known as endolysins. These enzymes hydrolyze the peptidoglycan (PG) layer of the host bacterial cell wall during the final stage of the phage replication cycle, leading to host cell lysis and the release of newly assembled phage particles (O’Flaherty, Ross & Coffey, 2009; Zermeño-Cervantes et al., 2023). In the present study, we identified two ORFs, ORF42 and ORF82, which encode putative endolysins, as well as two ORFs encoding Rz-like spanins, suggesting that the bacteriolytic activity of phage Pv27 involves these enzymes. It is well-established that endolysins degrade the peptidoglycan layer and inner membrane of the host cell, while the spanin complex subsequently targets and disrupts the outer membrane, completing the lysis process and ensuring efficient host cell destruction (Berry et al., 2012). In our analysis, we found that ORF42 shared 99.89% identity with the amino acid sequence of a predicted peptidase (family M15) from Vibrio phage phiKT1024 (OM249648.1), which is recognized as one of the main phage lytic enzymes (van Heijenoort, 2011). Similarly, ORF82 showed 97.92% identity with the amino acid sequence of a predicted chitinase from Vibrio phage phiKT1024 (OM249648.1). This chitinase, an endolysin with lysozyme activity, belongs to the glycoside hydrolase family 19. These findings suggest that phages Pv27 and phiKT1024 may share a similar lysis mechanism, utilizing endolysins and spanin complexes to efficiently lyse the host bacterium and release progeny phage particles.

Additionally, other annotated genes have been detected in the Pv27 genome, including a chaperonin GroEL (ORF24), which is known to prevent the misfolding or unfolding of polypeptides generated under stress conditions (Jaworek et al., 2020). Another identified gene, an HNH endonuclease (ORF62), is associated with the superinfection immunity of the phage, preventing interference from genetic material introduced from the external environment (Yu et al., 2018). Furthermore, a PhoH-like phosphate starvation-inducible protein (ORF299) was identified, which may function in phosphate metabolism and is generally overexpressed in bacteria during phosphate starvation (Isidro, Henriques & Tavares, 2004). Since phosphorus is a major element in nucleotide biosynthesis and a critical limiting macronutrient in the ocean, genes related to phosphorus acquisition, such as phoH, pstS, and phoA, may enhance Pv27’s ability to acquire phosphorus during viral infection (van Heijenoort, 2011).

Remarkably, phage Pv27 contains a large number of tRNA genes (23 tRNAs), comparable to phage phiTY18, which has 24 tRNAs (Liu et al., 2022). Previous study has shown that tRNAs are involved in translation and play essential roles in the phage infection process, particularly in the synthesis of capsid and tail proteins, critical components of the phage life cycle (Fu et al., 2023). Furthermore, virulent phages have been reported to encode more tRNAs than lysogenic phages, likely due to the significant correlation between tRNA distribution and codon usage, which enhances translational efficiency (Li et al., 2012). However, exceptions have been observed, such as lytic phages lacking tRNAs entirely (Li et al., 2023) or encoding only a single tRNA (Yang et al., 2020). The high number of tRNAs in Pv27 likely provides a selective advantage by supporting efficient protein synthesis, reducing reliance on the host’s translational machinery, and enabling Pv27 to maintain high replication efficiency and adaptability across diverse host conditions. Furthermore, no genes associated with antibiotic resistance, virulence, or lysogeny were detected in Pv27, indicating that it is a lytic and non-pathogenic phage. This finding is particularly significant, as phages carrying virulence or toxin genes can exacerbate bacterial infections and complicate treatment efforts (Droubogiannis & Katharios, 2022). These characteristics highlight phage Pv27 as a promising candidate for developing bioproducts to control and mitigate V. parahaemolyticus infections in shrimp aquaculture (Fu et al., 2023).

VIRIDIC and BLASTn analyses revealed that Pv27 shares the highest nucleotide homology with Vibrio phages, including phiKT1024, phiTY18, Va1, VB_VaC_TDDLMA, and VB_VaC_SRILMA. Phylogenetic tree analysis indicated that phage Pv27 and these phages belong to the same viral cluster within the Caudoviricetes class.

Our data also indicate that phage Pv27 effectively controls bacterial growth. However, the bacteriolytic activity curve shows a point where the bacterial population begins to recover from phage infection, although it does not reach the density of the control group. This phenomenon can be explained by the ability of bacteria to evolve resistance to phages over time, as observed in previous studies, which suggest that phage resistance is often associated with the emergence of anti-phage mutant bacterial strains (Castledine et al., 2022; Zeng et al., 2024). For instance, Zeng et al. (2024) reported that mutations within the flaG gene in the host genome can prevent phages from adsorbing to the bacterial cell surface. However, the development of such mutants often comes with trade-offs, including a reduced growth rate and altered carbon source utilization (Zeng et al., 2024), as well as decreased virulence and impaired biofilm formation (Castledine et al., 2022; Zeng et al., 2024). As a result, the recovered host bacterial population may exhibit a lower cell density than the untreated group.

Phage therapy is an innovative and sustainable alternative to antibiotics for managing bacterial diseases in shrimp aquaculture. Its high specificity, environmental friendliness, and compatibility with other biocontrol strategies make it a valuable tool for enhancing shrimp health. However, challenges such as bacterial resistance to phages, regulatory approvals, and environmental stability must be overcome for widespread implementation. With ongoing research and advancements in phage engineering, encapsulation techniques, phage therapy is poised to become a key solution for sustainable shrimp farming in the future.

Conclusions

In this study, phage Pv27 demonstrates significant potential as a biocontrol agent for V. parahaemolyticus infections in shrimp aquaculture. Its narrow host range, rapid replication, high lytic activity, and stability under various environmental conditions make it an effective candidate for use in shrimp farming. The genome of Pv27 reveals complex mechanisms for infection, including tail-related proteins and lytic enzymes, further supporting its efficacy in targeting V. parahaemolyticus. Notably, the lack of antibiotic resistance, virulence, or lysogeny genes in Pv27 genome enhances its safety profile for use in aquaculture. Overall, these findings underline the promising role of phage Pv27 in controlling bacterial pathogens in aquaculture, with potential for further development and integration into sustainable farming practices.

Supplemental Information

Supplemental Information 1 Putative ORFs in Vibrio phage Pv27 genome and their assigned functions.

A total of 267 ORFs which are predicted to encode for hypothetical proteins are omitted from the table.

Supplemental Information 2 Genome comparison of phage Pv27 with Vibrio phage phiKT1024, phiTY18, Va1 and VB_VaC_TDDMLA.

The shading below each genome indicates sequence similarities between the genomes, with different colors representing the levels of similarity.

Supplemental Information 3 Phylogenetic tree based on major capsid protein sequences .

The phylogenetic tree was constructed using the Maximum Likehood with 1000 bootstrap replicates. Reference sequences were selected based on the top-hit BLAST results of phage Pv27’s major capsid amino acid sequences.

Supplemental Information 4 Phylogenetic tree assembled based on terminase large subunit sequences.

The phylogenetic tree was constructed using the Maximum Likehood with 1000 bootstrap replicates. Reference sequences were selected based on the top-hit BLAST results of phage Pv27’s terminase large subunit amino acid sequences.

Supplemental Information 5 The fully assembled genome of phage Pv27.

Supplemental Information 6 Raw data.

Additional Information and Declarations

Competing Interests

The authors declare that they have no competing interests.

Author Contributions

Vu Thi Hien performed the experiments, analyzed the data, prepared figures and/or tables, authored or reviewed drafts of the article, and approved the final draft.

Pham Thi Lanh performed the experiments, analyzed the data, authored or reviewed drafts of the article, and approved the final draft.

Thao Thi Phuong Pham analyzed the data, authored or reviewed drafts of the article, and approved the final draft.

Khang Nam Tran analyzed the data, prepared figures and/or tables, and approved the final draft.

Nguyen Dinh Duy performed the experiments, prepared figures and/or tables, and approved the final draft.

Nguyen Thi Hoa performed the experiments, prepared figures and/or tables, and approved the final draft.

Nguyen Xuan Canh analyzed the data, authored or reviewed drafts of the article, and approved the final draft.

Quang Huy Nguyen analyzed the data, authored or reviewed drafts of the article, and approved the final draft.

Seil Kim analyzed the data, authored or reviewed drafts of the article, and approved the final draft.

Dong Van Quyen conceived and designed the experiments, analyzed the data, authored or reviewed drafts of the article, and approved the final draft.

DNA Deposition

The following information was supplied regarding the deposition of DNA sequences:

The genome sequence of phage Pv27 is available at NCBI: OR413349.

Data Availability

The following information was supplied regarding data availability:

The raw measurements are available in the Supplemental Files.

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
