# Peer review of "Isolation and characterization of a novel lytic bacteriophage Pv27 with biocontrol potential against Vibrio parahaemolyticus infections in shrimp"

_PeerJ, doi:10.7717/peerj.19421_

## Round 0.1 · original submission · Major Revisions

Two reviewers provided feedback on your manuscript, which I agree with, leading me to make the editorial decision of Major Revisions. They have raised a number of critical comments that must be addressed, specifically requiring release of the GenBank sequence so it can be thoroughly evaluated, reconciling some issues in the phylogenetic analyses, and adding additional details regarding the methodology and statistics, and adding additional references (and removing duplicates).

Reviewer 1 ·

Basic reporting

The phage genome has been deposited in GenBank; however, it is currently unavailable. The authors must request its release to ensure accessibility.

In lines 60–64, the authors focus solely on Vibrio parahaemolyticus as the causative agent of Acute Hepatopancreatic Necrosis Disease (AHPND). However, Vp can also cause other vibriosis infections that are not toxin-mediated, making bacteriophage-based control more feasible. Expanding this discussion would strengthen the manuscript.

Overall, the article is well-structured and follows a logical sequence. Most results are adequately described, and the discussion is sufficient. However, the authors could provide a broader discussion on the topic rather than focusing too narrowly on individual experimental tests.

Experimental design

The microbiological methodologies used for bacteriophage characterization are appropriate. However, the raw data only include the results of three replicates for the microbiological experiments. To ensure transparency and reproducibility, the authors should provide additional details, including the number of counted plaques, the dilution at which they were counted, and the volume inoculated. This information is essential to verify the accuracy of the results.

Regarding the statistical analysis, the manuscript mentions the software used but does not specify the statistical tests applied to determine significant differences. Instead, only p-values are reported. The authors should explicitly state the statistical tests used for each analysis.

Validity of the findings

While the phylogenetic analyses are correctly performed, according to ICTV guidelines, genus demarcation requires more than 70% genome-wide similarity, and species demarcation requires at least 95% similarity. Tools such as VIRIDIC should be used to conduct these necessary comparisons and accurately assess the taxonomic classification of the newly described phage.

The authors claim that the phage effectively controlled bacterial growth. However, the bacteriolytic activity curve shows a point where the bacterial population recovers from phage infection, although it does not reach the control group's density. The authors should clarify the criteria used to determine phage efficacy and further discuss the observed curve dynamics.

The authors repeatedly state that the phage clusters in a different subclade than phiTY18 and phiKT1024. However, this interpretation is incorrect. The VICTOR analysis algorithm groups these phages within the same species, making the claim that they belong to different subclades misleading. This misinterpretation should be corrected.

Additional comments

The manuscript has the potential for publication, but significant improvements are necessary in the aforementioned sections.

Reviewer 2 ·

Basic reporting

This manuscript describes the basic characteristics of the newly isolated Vibrio parahaemolyticus-infecting bacteriophage, Pv27. The authors isolated the phage and analyzed its morphological traits via TEM and its biological characteristics via a one-step growth curve, host challenge assay, etc. Furthermore, they assessed the genomic sequence, confirming the phage’s phylogenetic relationship with other V. parahaemolyticus-infecting phages.

Although the manuscript follows the typical narrative of bacteriophage research, the following issues were raised during the review process:
1. There is an insufficient number of references/examples for previously published V. parahaemolyticus phages. There are also reports describing V. parahaemolyticus phages with an icosahedral head and a long, flexible tail (Shipovirus type).
2. Similarly, please include the causative agents (toxins or virulence factors) responsible for AHPND in V. parahaemolyticus in the Introduction section.
3. There are duplicated references. Please check the references again (36 & 37).
4. Lines 140–143: Please provide full names if they appear for the first time in the text.
5. Lines 265–266: Please explain why the recovered host bacterial cells reached only a lower OD.
6. The genome sequence has not been released yet; therefore, the reviewer cannot verify the integrity of the genome analysis.
7. Please provide further explanation and discussion regarding the very narrow host range of phage Pv27.

Experimental design

1. Line 137: Please provide details about the TEM model used.
2. Is there a specific reason for using a 0.45 µm filter? It is not optimal for removing bacterial cells.

Validity of the findings

The genome sequence has not been released yet; therefore, the reviewer cannot verify the integrity of the genome analysis.

Additional comments

No comment.

---

## Round 0.2 · accepted · Accept

Thank you for providing the sequence in Genbank and carefully addressing the requested changes.

Reviewer 1 ·

Basic reporting

The authors addressed the observations made, releasing the sequence of the phage in Genbank so that it could be consulted, expanding the focus on other diseases caused by Vibrio parahaemolyticus, and broadening the discussion.

Experimental design

The authors addressed the observations made, to ensure adequate revision of the results by explicitly adding the raw data and placing the information corresponding to the statistical analyses.

Validity of the findings

With the inclusion of the taxonomic assignment by whole genome comparison and the discussion of the behavior observed in the bacteriolytic activity curve, I consider the findings described to be valid.

Additional comments

I have no further comments.